# Approach in inputs & outputs selection of Data Envelopment Analysis (DEA) efficiency measurement in hospitals: A systematic review

**M. Zulfakhar Zubir**[1,2], **A. Azimatun Noor** [2]*, **A. M. Mohd Rizal**[2], **A. Aziz Harith** [3], **M. Ihsanuddin Abas**[4], **Zuriyati Zakaria**[1], **Anwar Fazal A. Bakar**[2,5]

1 Medical Development Division, Ministry of Health Malaysia, Putrajaya, Malaysia, 2 Department of Public Health Medicine, Faculty of Medicine, Universiti Kebangsaan Malaysia, Kuala Lumpur, Malaysia, 3 Occupational and Aviation Medicine Department, University of Otago Wellington, Wellington, New Zealand, 4 Department of Public Health, Faculty of Medicine, Universiti Sultan Zainal Abidin, Terengganu, Malaysia, 5 Medical Practice Division, Ministry of Health Malaysia, Putrajaya, Malaysia

* azimatunnoor@ppukm.ukm.edu.my

**Data Availability Statement:** All relevant data are within the manuscript and its Supporting information files.

## Abstract

The efficiency and productivity evaluation process commonly employs Data Envelopment Analysis (DEA) as a performance tool in numerous fields, such as the healthcare industry (hospitals). Therefore, this review examined various hospital-based DEA articles involving input and output variable selection approaches and the recent DEA developments. The Preferred Reporting Items for Systematic Reviews and Meta-Analyses (PRISMA) methodology was utilised to extract 89 English articles containing empirical data between 2014 and 2022 from various databases (Web of Science, Scopus, PubMed, ScienceDirect, Springer Link, and Google Scholar). Furthermore, the DEA model parameters were determined using information from previous studies, while the approaches were identified narratively. This review grouped the approaches into four sections: literature review, data availability, systematic method, and expert judgement. An independent single strategy or a combination with other methods was then applied to these approaches. Consequently, the focus of this review on various methodologies employed in hospitals could limit its findings. Alternative approaches or techniques could be utilised to determine the input and output variables for a DEA analysis in a distinct area or based on different perspectives. The DEA application trend was also significantly similar to that of previous studies. Meanwhile, insufficient data was observed to support the usability of any DEA model in terms of fitting all model parameters. Therefore, several recommendations and methodological principles for DEA were proposed after analysing the existing literature.

## 1. Introduction

Efficiency is a well-established concept in the field of economics. Farrell proposed that efficiency measurement should consider all inputs and outputs, avoiding index number issues

**Funding:** The author(s) received no specific funding for this work.

**Competing interests:** The authors have declared that no competing interests exist.

and providing practical calculation methods. Hence, efficiency-related articles have attracted significant attention from various fields, including statisticians, economists, healthcare, and medicine [1]. Nevertheless, insufficient agreement regarding the optimal method is observed for measuring efficiency. For example, various methodologies are often used for efficiency-related articles in health facilities, including data envelopment analysis (DEA), stochastic frontier analysis (SFA), Pabon Lasso, and ratio analysis [2–4]. The World Health Organization (WHO) also introduced a unique approach to evaluating the effectiveness of healthcare systems in the Global Programme on Evidence for Health Policy Discussion Paper Series. Compared to a previous study in this field [5], this approach introduced numerous objectives of the healthcare system, including responsiveness (level and distribution), fair finance, health inequality, and the more traditional goal of improving population health.

Another report by the WHO evaluated the performance of the health system by assessing how well national health systems achieved three main objectives: good health, expectation responsiveness to the population, and fairness of financial contribution [6]. Despite the acknowledgement of this method, disagreement and criticism have occurred over the methodology employed. Conversely, a consensus has been observed regarding the importance of accurately directing these assessments, performing a more critical analysis, adopting a more constructive approach, and facilitating a crucial dialogue among stakeholders in the healthcare system [7–9]. Recently, the DEA has been used to compute the effectiveness of healthcare systems in 180 countries. This assessment is based on six key dimensions: clinical outcomes, health-adjusted life years, access, equity, safety, and resources [10]. Stakeholders must comprehend that universally applicable efficiency metrics for all healthcare systems are impossible. Therefore, a comprehensive understanding of the institutional arrangements, data, and measurements is necessary to select suitable measures, resources, and other health system components.

A framework is required following the analysis process. The optimal approach to implementing performance measurement is not to identify a minor adjustment as a supporting role to enhance one aspect of the health system outcomes. Instead, this identification should be utilised as a general strategy in gauging performance among the various system components [11, 12]. Numerous indicators, such as activity and expense comparison measures, are also available to assess whether limited health resources are utilised most efficiently. The primary focus of these indicators is based on quantitative metrics for evaluating hospital performance. Furthermore, the quality of hospital services can be examined using various indicators [13, 14]. Efficiency comparisons can also be assessed objectively using techniques from a solid economic theory. Currently, the DEA and SFA approaches are frequently applied to measure the efficiency of the healthcare industry [11]. Since the publication of Nunamaker's study, these strategies have been widely used in healthcare settings over the past 40 years [15–19]. Although the theoretical and methodological limitations have been acknowledged in DEA, this method has attracted interest from researchers who aim to address the limitations. Hence, these studies have developed multiple methods integrating DEA with other statistical techniques and methodologies to improve efficiency evaluation [20, 21].

## 1.1 DEA as an efficiency analysis tool in hospital

The DEA is a mathematical technique for assessing the relative efficiency of homogenous decision-making units (DMUs) with many input and output variables. Initially, this method was developed within operations research and econometrics. The effectiveness of a DMU is then evaluated concerning the effectiveness of each other members of the group. Nevertheless, one drawback of the DEA is its non-parametric and deterministic nature, suggesting that outliers

are more easily detected. Meanwhile, an efficient DMU usually involves maximum output production while utilising the same input levels as all other DMUs [17, 22]. Various DEA-related articles have proposed that this outcome is denoted as the Charnes, Cooper, and Rhodes (CCR) model or constant return to scale (CRS) assumption. This observation allows for examining input-output correlation without considering any congestion effects, indicating that the outputs can present a precise linear correlation with the inputs [10, 23].

Banker expanded the CCR model and the CRS assumption using a Banker, Charnes, and Cooper (BCC) model and variable returns to scale (VRS) assumption. This assumption suggests that the scales of the economies shifted with higher DMU size [23, 24]. The DEA approach also considers the model orientation (input or output-oriented) alongside the model type and returns to scale assumption. For example, a DMU in the input orientation assumption can control more inputs than outputs. Nonetheless, this statement can be argued that organisations can improve their outputs by utilising efficiency-oriented inputs [23, 25]. Hence, the input and output variables should be carefully considered when using the DEA to measure the effectiveness of a DMU or an organisation. This suggestion indicates that a precise, thorough, pertinent, and appropriate selection and combination of the input and output variables is necessary to effectively portray the functionality of a hospital while meeting the stakeholders' expectations and assessing its efficiency [18, 21]. Numerous advanced analyses have also been incorporated into DEA, such as the advanced CCR and BCC models, longitudinal or window analysis (Malmquist index), and statistical analysis (regression and bootstrapping methods) [20, 23, 25–27].

## 1.2 Input and output selections for hospital-based DEA applications

Multiple articles have demonstrated the practicality and potential of DEA in evaluating hospital efficiency [28–31]. Despite that DEA rating comparisons across several hospital-based articles produce helpful hypotheses, significant drawbacks are observed as follows:

1. The input and output metrics vary across different timeframes.

2. The DEA score distribution is highly skewed, rendering it inaccurate to rely on standard measures of central tendency.

3. The output metrics in the articles present significant divergence from each other.

4. The hospital production models and types possess substantial differences.

Certain hospital-based articles have reported that innovative strategies can provide valuable insights to decision-makers [23, 32]. These articles have also included the DEA for hospital-based applications. Generally, DEA-based applications involve health care performance measurement [15, 16, 18], categorisation or clustering of DEA techniques [20, 33], DEA comparison with other methods, countries or durations [28, 29, 30, 34], and development of novel knowledge and approaches concerning DEA assessment [17, 21]. Likewise, each stage in a systematic literature review (SLR) employs organised, transparent, and reproducible techniques to identify and integrate relevant articles to a particular topic comprehensively. The reviewer's methodology is meticulously recorded, allowing readers to track the decisions and actions taken and evaluate them [35]. Although numerous hospital-based DEA articles have been recorded, inadequate complete analysis has been observed. Consequently, this outcome requires further investigation, leading to a research gap involving hospital-based DEA articles.

This review investigated various hospital-based DEA articles for selecting the most suitable input and output variables. Notably, hospital institutions were chosen due to the significant challenges in assessing their efficiency. This limitation was further complicated by the dynamic

nature of service production and variation across several providers [25, 36, 37]. To the authors' knowledge, no reviews regarding hospital-based DEA articles involving optimal input and output variable selections were reported. Thus, this review addressed this research gap by observing the current trends in hospital-based DEA analyses. The remainder of this review is structured as follows: Section 2 describes the methodology used and the Preferred Reporting Items for Systematic Reviews and Meta-Analyses (PRISMA) statements. Section 3 presents the literature review of the relevant articles and their corresponding discussions concerning the input and output variable selections for hospital-based DEA applications. Finally, Section 5 highlights the limitations and conclusions of this review.

## 2. Methodology

This section discusses the methodology used to obtain relevant hospital-based DEA articles. The PRISMA methodology involved Web of Science, Scopus, PubMed, ScienceDirect, Springer Link, and Google Scholar databases. This process conducted the SLR, eligibility with exclusion criteria, review stages (identification, screening, and eligibility), and data abstraction with analysis.

### 2.1 PRISMA

The PRISMA methodology concisely collects components for documenting SLRs and meta-analyses based on supported evidence. Even though this approach typically focuses on reporting reviews evaluating intervention effects, it can also be used as a basis for publishing SLRs with objectives other than assessing interventions (Appendices A and B) [38]. Hence, a comprehensive manual on the SLR methodological approach is required for future researchers. This SLR initiates with developing and verifying the review method, publication standard, and reporting standard or guidance. These articles can then provide a systematic guideline for researchers outlining the factors necessitating consideration during the review process [39].

### 2.2 Journal databases

Various articles published from 2014 until 2022 were obtained on 5th April 2023, using six databases: Web of Science, Scopus, PubMed, ScienceDirect, Springer Link, and Google Scholar. The analysis of the search engines revealed significant performance discrepancies, indicating the absence of an optimal search approach. Therefore, searchers must be well-trained, capable of evaluating the strengths and weaknesses of a system, and able to determine where and how to search based on that information to use them effectively. The six databases were selected based on their potential to provide a meticulously curated medical database with recall-enhancing features, tools, and other alternatives to optimise precision [40, 41].

### 2.3 Identification

This systematic review process comprised four stages (identification, screening, quality appraisal, analysis). Several search terms were identified during the first stage, which involved searching previous articles using various terms: "efficiency*", "performance*", "productivity*", "benchmark*", "hospital*", "data envelopment analysis", and "DEA". The search string was modified according to the requirement of database. The records were exported from the databases into Microsoft Excel sheet for screening. The final query string is as follow:

((((("hospital") AND ("efficiency")) OR (("hospital") AND ("performance")) OR (("hospital") AND ("benchmark")) OR (("hospital") AND ("productivity"))) AND (("data envelopment analysis") OR ("DEA")))

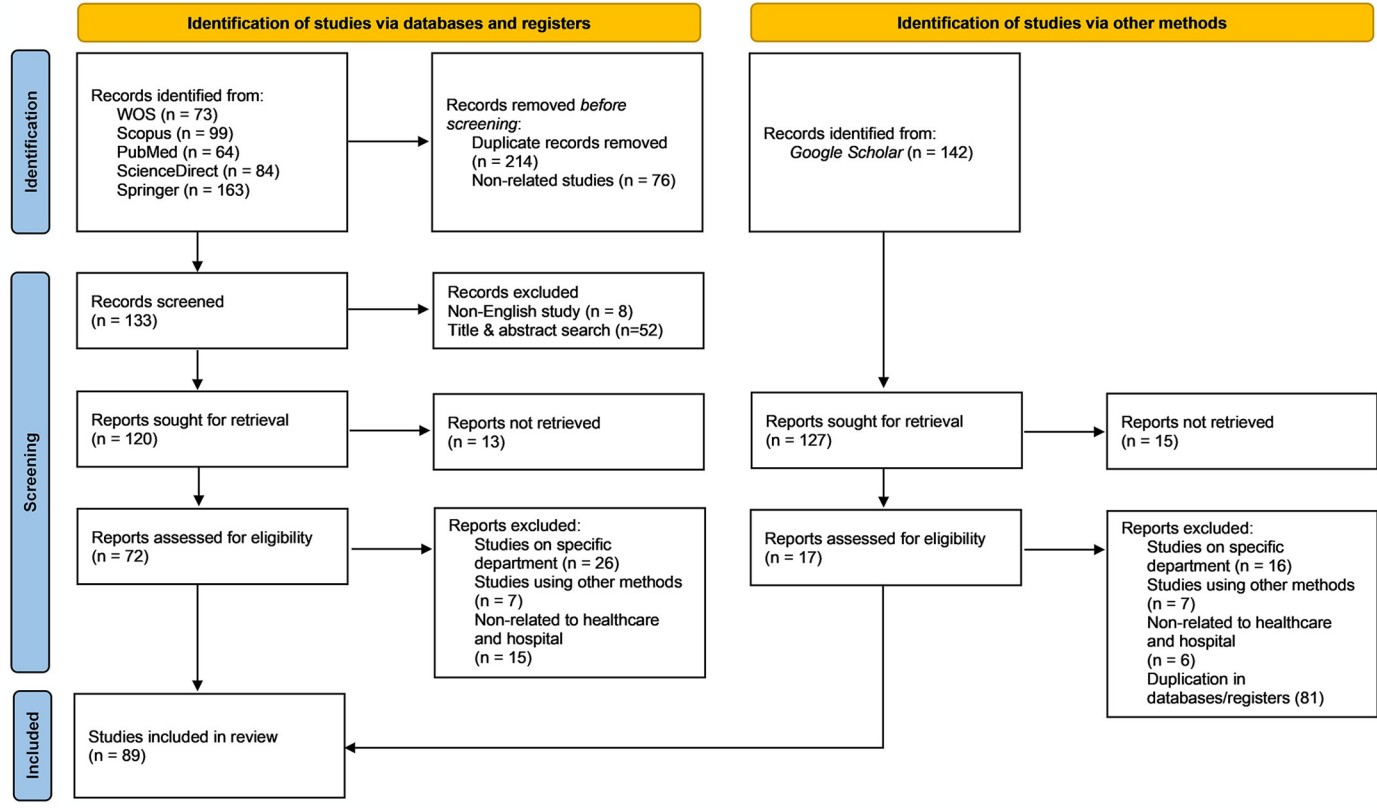

**Fig 1. The PRISMA diagram of the search process.**

## 2.4 Screening

The inclusion and exclusion criteria were established in this review. The titles and abstracts were independently screened by three reviewers. Only articles containing empirical data were initially selected, and this process excluded review articles (SLR and SR), book series, books, book chapters, and conference proceedings. Non-English articles were then excluded in the search attempts, avoiding any ambiguity or difficulty in translation. Subsequently, a nine-year duration was chosen for the chronology (2014–2022) to observe significant developments in research and relevant articles. This duration also functioned as a continuation of a previous study by O'Neill *et al.* (1984–2004), Cantor and Poh (1994–2017), and Kohl *et al.* (2005–2016). Consequently, 89 articles were finalised for the quality appraisal stage. Fig 1 depicts the PRISMA diagram, which provides a detailed description of the entire search procedure.

## 2.5 Quality appraisal

A quality appraisal stage was conducted to ensure that the methodology and analysis of the selected articles were performed satisfactorily. This process contained two quality appraisal tools: knowledge transfer [29, 42] and economic evaluations and efficiency measurement [43, 44]. Mitton *et al.* developed a 15-point scale that covered several topics: literature evaluation, research gap identification, question, design, validity and reliability, data collection, population, sampling, and result analysis and report. These criteria were evaluated using a score range between 0 and 3: 0 for not being present or reported, 1 for being present but of low quality, 2 for being present and mid-range quality, or 3 for being present and of high quality [42].

Another checklist by Varabyova and Müller employed four dimensions: reporting, external validity, bias, and power. All items on the quality assessment checklist were assigned a score of either 0 (indicating no or unclear) or 1 (indicating yes). One item in the checklist also focused on conducting a second-stage analysis to investigate potential sources of bias in the study. The articles with and without second-stage analysis received maximum scores of 14 and 13, respectively. This checklist assessed the article from an economic perspective to ensure the findings could be used in policy analysis and managerial decisions. Only the items relevant to the design of the article were utilised to establish the maximum score (100%) for each study [44]. Overall, no recognised standards for assessing the planning or implementation of research on healthcare efficiency indicators were recorded. Thus, the scientific soundness of the chosen article was investigated using two tools to improve robustness and minimise bias. Two co-authors from different institutions evaluated each selected article separately using both tools to enhance reliability. A third reviewer was then requested to assess an article if a disagreement occurred.

## 2.6 Data extraction and analysis

The selected articles were subjected to examination and analysis. Specific articles were also prioritised to meet the objectives directly. The data extraction process could be performed by reading the abstracts and the entire article. Meanwhile, content and quantitative and qualitative analyses were used to determine the input and output selection approaches for the hospital-based DEA articles. Four reviewers extracted the data independently using a standardized data extraction form which is organized using Microsoft Excel. The information in this form included publication year, country of study, studied hospital type, number of hospitals, number of observations (DMUs), model type, returns to scale, model orientation, measured efficiency type, input, output, number of models, application of second stage analysis, and approaches used in selecting input or output variables.

## 2.7 Statistical analysis

In evaluating the studies, the intra-class correlation (ICC) was used to measure the agreement between two raters (co-authors). This process examined the dependability of ratings by comparing the variability between various evaluations of the same subject regarding the overall variation observed across all ratings and subjects. Each of the evaluation processes was also quantitative. Meanwhile, the ICC coefficient values for Mitton *et al.*'s (15-point scale) and Varabyova and Müller's (economic evaluation and efficiency measurement) studies were 0.956 and 0.984, respectively. No articles were excluded at this stage as the review encompassed qualitative and quantitative aspects. Nonetheless, highly rated articles were considered highly in the data analysis and result interpretation processes.

## 3. Results

All the 89 articles included in this analysis were retrospective studies published between 2014 and 2022. Appendices C and D contain a comprehensive summary of all the selected articles.

### 3.1 Efficiency analysis

The efficiency analysis in DEA primarily focused on the data by quantifying the set performance of DMUs. Given that the definition of DMU is generic and broad, this review focused on "hospital". Typically, the four main efficiency concepts are technical, scale, pricing, and allocative efficiencies [25]. Certain studies also described efficiency as technical, pure, scale,

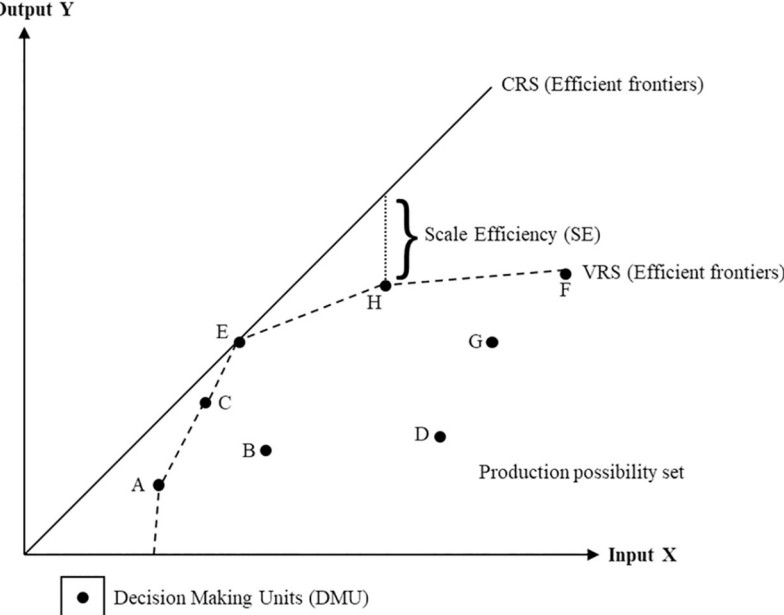

**Fig 2. The efficiency measurement concepts in DEA.**

allocative, cost, and congestion. The DEA could perform efficiency analysis at a single point in time and over time [26]. Thus, the data could be categorised as cross-sectional (single period) or longitudinal (panel data). Specifically, the longitudinal analysis of DEA utilised two approaches to quantify efficiency: the Malmquist Productivity Index (MPI) and Window Analysis (WA).

Out of the 89 articles, a significant portion of them (32.58%, 29 of 89) focused only on evaluating hospital performance using Pure Technical Efficiency (PTE) [45–73]. This metric is defined as the effectiveness of an input set producing an output on the VRS frontier [49, 68]. Hence, a hospital is deemed technically efficient when it generates the highest quantity of outputs using the fewest inputs. The overall Technical Efficiency (TE) is determined by multiplying the Scale Efficiency (SE) and PTE [74, 75]. Generally, TE refers to the efficiency measured under the CRS production frontier. In contrast, SE measures how much a unit deviates from an optimal scale, which is an area involving CRS in the correlation between outputs and inputs [76, 77]. Hence, the equation for TE is expressed as follows:

$$TE\,(\theta_{CCR}) = PTE\,(\theta_{BCC})\,\times SE$$

Fig 2 provides a geometric representation of the concepts involved in efficiency measurement using the DEA. Of the 89 articles, 26 (29.21%) measured the overall TE [76–101]. Additionally, 24 (26.97%) computed the TE, PTE, and SE [74, 75, 102–125]. Even though the remaining articles were assessed by combining TE and PTE, certain articles did not explicitly specify the tested efficiency type (see S5 Appendix).

### 3.2 Model parameters

The DEA was applied using four considerations specified by the researcher: model type, technological assumption of the delivery process, model orientation, and input-output combination [112, 121]. This model could be further analysed or extended through a second stage or

integrated with other statistical methods. Consequently, this process could improve efficiency measurement, understanding of the variation or difference in organisational performance, and evaluation of the productivity of the organisation over a specific period [91, 110, 119]. Considering that the performance was analysed over a certain period, the data type was also essential.

**3.2.1 Model type.** The DEA has been utilised to assess the performance of various entities involved in diverse activities under different circumstances. This process leads to numerous models and extensions explaining the intricate and frequently unpredictable correlations between multiple inputs and outputs in organisation activities or productions [106, 124]. Hence, these models can be described as basic DEA and extension models. Certain articles have also denoted the model as Radial, Non-radial and Oriented, Non-radial and Non-oriented, and Radial and Non-radial [23, 125]. Most articles in this review (80.90%, 72 of 89) used Radial DEA models [45, 47–51, 53–61, 64–76, 78, 79, 81–85, 88–93, 95–99, 101–115, 117–123, 126–129].

The BCC, CCR, or a mixture of both models were used to measure efficiency by examining the radial changes in input and output values. Nevertheless, only 7.87% (7 of 89) [46, 52, 62, 63, 94, 100, 130] or 4.49% (4 of 89) [80, 86, 87, 124] employed Non-radial and Oriented or Non-radial and Non-oriented models, respectively. The Non-radial model deviated from the conventional approach of proportional input or output changes and instead focused on addressing slacks directly. Only one article was observed using the Radial and Non-radial models [77], while one combined Radial, Non-radial, and Oriented models to measure efficiency [116]. The remaining four articles did not explicitly specify the model employed in the study (see S6 Appendix) [131–134].

**3.2.2 Model orientation.** Orientation refers to the specific direction in which input or output is measured to determine efficiency. The primary evaluation objective is to either increase output or decrease input. Most articles in this review (55.06%, 49 of 89) applied input-orientated DEA models [45–47, 49–53, 57, 59, 63–68, 70–72, 74–76, 79, 90–93, 95, 97, 98, 102–106, 108–110, 112–115, 117, 120, 123, 126, 127, 129, 134]. These articles selected the input orientation to align with the standard practice in healthcare facilities of minimising inputs while achieving a desired output level. Thus, the organisation acquired minimal or non-existent authority over the output [45, 50, 109].

Approximately 25.84% (23 of 89) of the articles presented contradictory findings [48, 54–56, 58, 60, 61, 69, 73, 78, 81–85, 101, 107, 111, 116, 118, 121, 122, 133]. Given the fixed and non-flexible nature of the input, the organisation should strive to raise its output. This outcome implied that output-orientated DEA models were more appropriate in their respective settings [78, 82, 83]. Meanwhile, only 5.62% (5 of 89) [80, 86, 87, 99, 124] or 3.37% (3 of 89) [94, 100, 128] employed non-orientated or combined input and output-orientated DEA models, respectively. The remaining articles did not specify the orientation used in their measurements and did not clearly state their orientation (see S7 Appendix) [62, 77, 88, 89, 96, 119, 130–132].

**3.2.3 Returns to scale assumption.** Approximately one-third of the articles (35.96%, 32 of 89) involving the returns to scale assumption combined CRS and VRS assumptions in evaluating efficiency [74, 75, 78, 91, 101–124, 126–129]. These articles compared the efficiency score to acquire a more comprehensive understanding of the organisation. Moreover, these articles provided additional knowledge on how they might utilise each assumption to enhance hospital services [101, 108, 113]. Another one-third of the articles (32.58%, 29 of 89) used the VRS assumption and implied a significant correlation between the outputs of organisations (DMUs) (increase or decrease) and inputs [45–73]. Likewise, 20.22% (18 of 89) [76, 79, 81–85, 88–90, 92, 93, 95–100] assumed that the outputs of their organisations (DMUs) varied (increase or decrease) similarly to the inputs (see S8 Appendix).

**3.2.4 Input and output selections.** Appropriate input and output selections are necessary for conducting a comprehensive efficiency evaluation. Therefore, identifying the key attributes depicting the investigated process or output is critical. This process implies that all relevant resources should be incorporated into the inputs, while the administrative objectives of the organisations (DMUs) should be outlined in the outputs [52, 104]. Nonetheless, suitable inputs and outputs can present varying features depending on the situation. Data availability also requires significant consideration alongside appropriate input and output selections. Hence, various recommendations have been presented involving locating suitable measures [76, 131, 133]. This process is further discussed as the main objective of this review.

Several articles employed input and output classifications for measuring efficiency, including capacity, labour, and expenses-related or capital investment, labour, and operating expenses. Specific articles also further delineated this classification process into sub-categories. For example, the outputs were classified as inpatient with outpatient services and effectiveness (quality). Other outputs were classified into two categories: activity (inpatient and outpatient) and quality-related (effectiveness dimension) [20, 21, 25, 32]. Table 1 lists the input and output classification and sub-classification processes in this review. Tables 2 and 3 summarise the details of each sub-classification frequency distribution and percentages.

*3.2.4.1 Capacity-related inputs.* The size, capacity, and functioning of a hospital as a health service are determined mainly by its number of fully staffed and operating beds. Out of the 89 articles, 75 of them (84.27%) considered the number of beds (general, intensive care unit and special) as inputs in their analyses [47–54, 56–61, 63–73, 75, 77–92, 94–100, 102–108, 111–115, 117, 118, 120–124, 127–130, 132–134]. Only seven (9.33%) of the 75 articles used bed-related data as their inputs (bed type, cost, or ratio) [60, 65, 70, 77, 97, 100, 128]. Another 12 (16.00%) of the 75 articles studies combined beds and capital assets as capacity-related inputs [69, 77, 84, 88, 89, 100, 102, 105, 108, 121, 124, 132]. Even though only one article employed capital assets as its input, it was combined with cost-related assets. Unlike other articles, it became apparent why this article did not include beds as part of its input [55]. Overall, the primary capacity-related input in these articles was the number of general beds. This input was followed by the number of facility types and the number of medical equipment.

**Table 1. Summary of the used input and output categories.**

| Input | Percentage (%) | Output | Percentage (%) |
|---|---|---|---|
| Capacity-related: | 30.06 | Production-related | 94.30 |
| Beds | 24.85 | Inpatients | 42.62 |
| Capital assets | 5.21 | Outpatients | 24.83 |
| | | Adjusted scores | 13.42 |
| Cost-related: | 15.34 | Combination | 12.08 |
| Total costs | 7.06 | Monetary | 4.70 |
| Medication & service costs | 5.21 | Imaging and Laboratory | 2.35 |
| Labour costs | 2.15 | | |
| Equipment costs | 0.92 | Quality-related | 5.70 |
| | | Patients | 72.22 |
| Staff-related: | 52.76 | Staff | 27.78 |
| Doctors | 17.18 | | |
| Nurses | 13.19 | | |
| Clinical staff | 10.74 | | |
| Non-clinical staff | 7.98 | | |
| Combination | 3.68 | | |
| **Other specific outputs:** | 1.84 | | |

**Table 2. Complete list summary of the used inputs.**

| Input | Frequency | Percentage(%) |
|---|---|---|
| Capacity-related: | | |
| Number of general beds | 73 | 74.49 |
| Number of facility types (area, space) | 10 | 10.20 |
| Number of medical equipment | 5 | 5.10 |
| Number of acute beds | 4 | 4.08 |
| Adjusted bed value (ratio, log value) | 4 | 4.08 |
| Number of total assets | 2 | 2.04 |
| Cost-related: | | |
| Total operating costs | 14 | 28 |
| Fixed costs | 7 | 14 |
| Service costs | 5 | 10 |
| Consumable costs | 5 | 10 |
| Total labour costs | 4 | 8 |
| Combination | 4 | 8 |
| Medication costs | 3 | 6 |
| Clinical staff costs | 2 | 4 |
| Capital costs | 2 | 4 |
| Beds cost | 2 | 4 |
| Non-clinical staff costs | 1 | 2 |
| Medical equipment costs | 1 | 2 |
| Staff-related: | | |
| Number of specialist, physician, doctor, GP, dentist | 38 | 22.09 |
| Number of nurses, midwives, nursing staff | 30 | 17.44 |
| Number of medical staff | 23 | 13.37 |
| Number of non-clinical staff | 18 | 10.47 |
| Full-time equivalent specialist, physician, doctor, GP, dentist | 16 | 9.30 |
| Full-time equivalent nurses, midwives, nursing staff | 11 | 6.40 |
| Number of combinations of staff | 9 | 5.23 |
| Full-time equivalent non-clinical staff | 8 | 4.65 |
| Full-time equivalent medical staff | 6 | 3.49 |
| Number of allied health staff | 5 | 2.91 |
| Ratio of doctors | 2 | 1.16 |
| Ratio of nurses | 2 | 1.16 |
| Full-time equivalent combination of staff | 2 | 1.16 |
| Number of combinations of medical staff | 1 | 0.58 |
| Combination of staff log value | 1 | 0.58 |
| Others: | | |
| Number of admissions | 1 | 16.67 |
| Average length of stay | 1 | 16.67 |
| Annual revenue | 1 | 16.67 |
| Discharge log value | 1 | 16.67 |
| Inpatient discharge rate | 1 | 16.67 |
| Population | 1 | 16.67 |

*3.2.4.2 Cost-related inputs.* Cost-related input was the least utilised in all the articles. Of the 89 articles, only 31 (34.83%) were applicable. Another three of the 31 articles specifically used cost-related inputs [62, 101, 119]. Interestingly, most of these 31 articles (90.32%) combined capacity and staff-related inputs in their analyses [45–47, 50, 55, 61–63, 69, 74–78, 82, 90, 93,

**Table 3. Complete list summary of the used outputs.**

| Output | Frequency | Percentage(%) |
|---|---|---|
| Production-related: | | |
| Number of outpatients | 50 | 16.78 |
| Number of Inpatients (admission and discharge) | 33 | 11.07 |
| Total number of operations | 28 | 9.40 |
| Number of inpatients | 23 | 7.72 |
| Number of emergency outpatients | 22 | 7.38 |
| Number of inpatient days | 14 | 4.70 |
| Bed occupancy rate (BOR) | 12 | 4.03 |
| Inpatient adjusted value (Casemix, price, and ratio) | 12 | 4.03 |
| Discharge adjusted value (Casemix) | 11 | 3.69 |
| Number of general & emergency outpatients | 10 | 3.36 |
| Average length of stay (ALOS) | 9 | 3.02 |
| Total revenue | 8 | 2.68 |
| Ratio adjusted value | 7 | 2.35 |
| Number of daycare patients | 6 | 2.01 |
| Number of birth deliveries (normal and caesarean) | 6 | 2.01 |
| Number of laboratory & radiology (services and examinations) | 5 | 1.68 |
| Number of operations | 4 | 1.34 |
| Outpatient adjusted value (Casemix, price, and ratio) | 4 | 1.34 |
| Number of total patients | 3 | 1.01 |
| Number of family medicine outpatients | 2 | 0.67 |
| Number of obstetric outpatients (ANC and PNC) | 2 | 0.67 |
| Number of emergency outpatient surgeries | 2 | 0.67 |
| Bed turnover rate (BTR) | 2 | 0.67 |
| Bed turn over interval (TOI) | 2 | 0.67 |
| Number of surgeries (minor and major) | 2 | 0.67 |
| Log adjusted value | 2 | 0.67 |
| Score adjusted value | 2 | 0.67 |
| Operating income | 2 | 0.67 |
| Inpatient income | 2 | 0.67 |
| Number of diagnostics (visits and procedures) | 2 | 0.67 |
| Number of medical outpatients | 1 | 0.34 |
| Number of allied health outpatients | 1 | 0.34 |
| Average number of admissions and discharge | 1 | 0.34 |
| Number of special patients | 1 | 0.34 |
| Number of total examinations | 1 | 0.34 |
| Discharge adjusted value (Casemix) | 1 | 0.34 |
| Death adjusted value (Casemix) | 1 | 0.34 |
| Outpatient revenue | 1 | 0.34 |
| Examination revenue | 1 | 0.34 |
| Quality-related: | | |
| Mortality rate (infant, adult, and specific diseases) | 9 | 50.00 |
| Revisit rate (outpatient and emergency) | 3 | 16.67 |
| Number of students | 2 | 11.11 |
| Patient's satisfaction score | 1 | 5.56 |
| Staff's satisfaction score | 1 | 5.56 |
| Number of medical inquiries | 1 | 5.56 |
| Management's score | 1 | 5.56 |

94, 100, 101, 108, 110, 111, 113, 115, 119, 124, 128, 129, 131, 132]. Overall, the primary cost-related input utilised in these articles was total operational cost, followed by fixed costs. Subsequently, service and consumable costs followed behind.

*3.2.4.3 Staff-related inputs.* Most articles (93.26%, 83 of 89) employed staff-related inputs [45–54, 56–61, 63–89, 91–118, 120–124, 126–130, 133, 134]. Another two of the 83 articles combined the number of staff (staff-related) and labour cost (cost-related) as their inputs [74, 77]. Alternatively, cost-related inputs (labour or operating costs) substituted staff-related input as proxies in six articles [55, 62, 90, 119, 131, 132]. The staff-related input values exhibited variability across the observed articles, while most articles employed arithmetic numbers (actual). A full-time equivalent and a ratio of specific values followed this input. Overall, these articles demonstrated that the number of doctors was the most common input, followed by the number of nurses and clinical staff.

*3.2.4.4 Production-related outputs.* A significant portion (98.88%, 88 of 89) of the articles highlighted production-related outputs [45–91, 93–124, 126–134]. Only eight (8 of 89) articles combined production and quality-related outputs [60, 72, 83, 86, 94, 97, 127, 132]. The most prevalent production-related output in these articles was the number of outpatients. This output was sequentially followed by the number of inpatients (admission and discharge), the total number of operations, and the number of inpatients.

*3.2.4.5 Quality-related outputs.* Quality-related outputs were less prominent than production-related outputs, and only nine (9 of 89) articles employed quality-related outputs [60, 72, 83, 86, 92, 94, 97, 127, 132]. Notably, one article focused exclusively on a quality-related output in their research, aligning with the objectives of the study [92]. Overall, the mortality rate (infant, adult, and specific diseases) was the most applied quality-related output. This output was followed by revisit rates (outpatients and emergency) and the number of students.

**3.2.5 Extended analysis and data type.** Approximately 80 articles (89.89%, 80 of 89) conducted extended analysis in their analyses [45–55, 57–63, 65–68, 70–76, 78–93, 95–97, 100–115, 117–124, 127–134]. The applied data type was also almost equally distributed. Among them, 51 articles (57.30%, 51 of 89) [46, 48, 51, 57–59, 66, 69–72, 74–76, 78, 79, 81–84, 87, 90, 91, 93, 95, 96, 99, 100, 102, 104–108, 110, 111, 113–115, 119–121, 123, 124, 126, 128–134] used panel data, In contrast, 38 articles (42.70%, 38 of 89) [45, 47, 49, 50, 52–56, 60–65, 67, 68, 73, 77, 80, 81, 85, 86, 88, 89, 92, 94, 97, 98, 101, 103, 109, 112, 116–118, 122, 127] employed cross-sectional data in their investigations.

Forty extended analyses were identified within the included articles, in which one or multiple extended analyses were used for each study (some mentioned as "stages"). Out of the extended analysis-related 80 articles, 46 integrated two or more extended analysis in their DEA measurements [45, 46, 50–53, 57, 59, 62, 63, 66, 67, 70, 72–74, 76, 78, 79, 82–85, 87, 90–93, 95, 100, 101, 106–108, 113–115, 118–120, 127, 129, 130, 132–134]. The maximum number of extended analyses in all 80 articles were five [66, 91, 93], in which regression analysis (29.11%, 46 of 158) was primarily used for assessing hospital efficiency. This analysis type was sequentially followed by production function analysis (16.46%, 26 of 158), statistical analysis (15.82%, 25 of 158) and resampling methods (15.82%, 25 of 15). Table 4 tabulates the complete list of the specific analyses for each classification (see S4 Appendix).

## 3.3 Input and output selection approaches

Various approaches or methods were adopted by the 89 articles in selecting inputs and outputs for hospital-based DEA. Each article relied on previous studies or literature reviews as the main or partial component of their methodology for selecting input and result variables. Only

**Table 4. Complete list summary of extended analyses.**

| Extended analysis | N | % | Extended analysis | N | % |
|---|---|---|---|---|---|
| Regression analysis | 46 | 29.11 | Production function analysis | 26 | 16.46 |
| Tobit regression | 26 | 16.46 | Malmquist productivity index | 21 | 13.29 |
| Ordinary least squares regression | 11 | 6.96 | Window analysis | 3 | 1.90 |
| Truncated regression | 5 | 3.16 | GML[a] Index | 1 | 0.63 |
| Generalised estimating equations regression | 1 | 0.63 | gMMP[b] Index | 1 | 0.63 |
| Linear regression | 1 | 0.63 | | | |
| Multinomial Logit regression | 1 | 0.63 | Resampling methods | 25 | 15.82 |
| Beta regression | 1 | 0.63 | Bootstrap method | 24 | 15.19 |
| | | | Monte Carlo simulation | 1 | 0.63 |
| Statistical analysis | 25 | 15.82 | | | |
| Mann—Whitney U test | 5 | 3.16 | Performance measurements | 13 | 8.23 |
| Wilcoxon signed-rank test | 4 | 2.53 | Benchmarking method | 7 | 4.43 |
| Kruskal-Wallis test | 4 | 2.53 | Stochastic frontier analysis | 2 | 1.27 |
| F-test | 2 | 1.27 | Isodata analysis | 1 | 0.63 |
| T-test | 2 | 1.27 | Super efficiency analysis | 1 | 0.63 |
| Paired T-test | 1 | 0.63 | Fitting adjustment | 1 | 0.63 |
| Central Limit Theorem | 1 | 0.63 | Pabon Lasso technique | 1 | 0.63 |
| Li-test | 1 | 0.63 | | | |
| Chi-square test | 1 | 0.63 | Correlation analysis | 11 | 6.96 |
| Repeated measures ANOVA | 1 | 0.63 | Spearman's rank correlation | 8 | 5.06 |
| Theil index | 1 | 0.63 | Pearson's correlation | 2 | 1.27 |
| Unpaired T-test | 1 | 0.63 | Grey's correlation | 1 | 0.63 |
| Univariate analysis | 1 | 0.63 | | | |
| | | | Clustering analysis | 8 | 5.06 |
| Matching methods | 3 | 1.90 | Cluster analysis | 6 | 3.80 |
| Propensity score matching | 3 | 1.90 | k-means clustering | 1 | 0.63 |
| | | | MST-kNN clustering algorithm | 1 | 0.63 |

Note:

[a] global Malmquist-Luenberger;

[b] generalised meta frontier Malmquist productivity

a few articles explicitly indicated using a local DEA efficiency study from their respective country as the reference for input and output selections. Certain articles also employed a combination of methodologies. Meanwhile, 63 of the 89 articles (70.79%) utilised only literature review to determine the input and output variables [45–47, 49–54, 58, 59, 61, 63–77, 80, 81, 84, 85, 88–91, 93, 95–97, 100, 102, 104, 107, 109, 110, 112, 114–121, 123, 124, 126, 127, 129, 131–134]. Meanwhile, the remaining articles employed a literature review in combination with other approaches, highlighting diverse combinations. Nonetheless, the most prevalent combination approach identified was a literature review combined with data availability (13.48%, 12 of 89) [55, 56, 78, 82, 103, 105, 106, 111, 113, 122, 128, 130]. This combination was followed by the literature review with systematic method (5.62%, 5 of 89) [57, 60, 83, 101, 108] and the literature review with DMU limitation (5.62%, 5 of 89) [48, 79, 86, 94, 98]. A maximum combination of four was also observed in one article [87]. Table 5 lists the complete list of various specific approaches.

**Table 5. Classification summary of the input and output selection approaches.**

| Approaches | N | % |
|---|---|---|
| Literature review | 63 | 70.79 |
| Literature review | 60 | 67.42 |
| Local studies literature review | 3 | 3.37 |
| Literature review and data availability | 12 | 13.48 |
| Literature review and data availability | 11 | 12.36 |
| Local studies literature review and data availability | 1 | 1.12 |
| Literature review and systematic method | 5 | 5.62 |
| Literature review and Delphi method | 2 | 2.25 |
| Literature review and Promethee method | 1 | 1.12 |
| Literature review and bibliometric analysis | 1 | 1.12 |
| Literature review and variance filter analysis | 1 | 1.12 |
| Literature review and DMU limitation | 5 | 5.62 |
| Literature review and expert judgement | 3 | 3.37 |
| Literature review, data availability, expert judgement, and DMU limitation | 1 | 1.12 |

## 4. Discussions

The size of the DEA universe can be intimidating to unfamiliar individuals. Even when the literature is limited to healthcare applications, reading every previous study to acquire knowledge from their experiences is exceedingly challenging. Consequently, the 89 articles were subjected to meticulous examination to accomplish the objectives of this review. Nunamaker was the first to publish a health application involving the DEA to examine nursing services. Subsequently, Sherman released a second DEA article evaluating the medical and surgical departments of seven hospitals [135, 136]. Hence, these articles have evolved DEA applications in the healthcare industry over the past four decades. The quality of the articles has advanced due to access to resources and information technology [23, 25, 137, 138].

### 4.1 Researchers' input and output selection approaches involving DEA for hospital efficiency measurement

The relative effectiveness of various institutions, including businesses, hospitals, universities, and government agencies, is frequently assessed using DEA. Conversely, these assessments are different from traditional-based analyses. Providing healthcare services in hospital-based environments is also distinct from the manufacturing process. Raw materials undergo a physical transformation to become final commodities in a conventional factory, in which participation and co-production are absent due to the exclusion of the customer component. Therefore, identifying the appropriate variables is difficult due to the involvement of patients in the process. The effectiveness (quality component) in healthcare is also equally crucial alongside performance and efficiency [86, 92, 127]. Even though DEA studies have not highlighted a standard set of input and output, several guidelines have been recommended using analytic procedures or principles to aid the optimal variable selection process [139–143].

**4.1.1 Literature review.** A literature review remains a commonly employed method and is often regarded as one of the most effective techniques to place a study within the body of knowledge. This method contains numerous review types (narrative, rapid, scoping, or systematic reviews) functioning as foundations or building blocks for knowledge advancement, theory development, and improvement area identifications [144–146]. The literature reviews examined in this study were also the most prevalent approach for input and output selections

concerning hospital efficiency-based DEA analyses. This review revealed that all the articles utilised literature review either as the primary method or as part of a combined approach.

None of the articles provided detailed information about their literature review approaches. Nevertheless, few articles explicitly mentioned selecting literature from their local country to compare their findings with previous local studies [49, 85, 104, 124]. Typically, the DEA is a non-parametric technique relying entirely on the observed input-output combinations of the sampled units. This process does not necessitate any presumptions regarding the functional structure correlations between inputs and outputs [76, 114]. Given that the DEA could measure the efficiency value depending on the objectives (even if it yielded a less significant value), an advantage was observed for these articles involving input and output selections based on the literature review [139, 142]. Even though this method remained valid for academicians, other assessors (managers, economists or policymakers) could perceive it as contradictory to their practical perspectives. Hence, several factors must be considered from these individuals' perspectives, including different indicators, production objectives, and policies.

**4.1.2 Data availability.** The DEA relies on the homogeneity of the assessment of a unit. The DMU is presumed to produce comparable activities or products using resources and technology in the same environment. Therefore, a common set of similar inputs and outputs can be established. Certain factors also require consideration when large hospital-related datasets are involved, including data quality, availability, scale, and type [139, 141]. Although this review indicated that the examined articles used literature reviews to select the input and output variables, this selection method depended on data availability. These articles only developed a few solutions to address the limitation. For example, few articles exclusively gathered data available within their scopes [87, 103, 130]. Certain articles also omitted the DMUs with incomplete data, focusing their analyses on DMUs with complete data [78, 105]. Thus, the DEA was advantageous in measuring only the relative efficiency or the production frontier of the units included in the analysis. Specific articles also applied the DEA during a defined data availability period to ensure all necessary input and output variables were complete [55, 113]. Although the DEA with missing incomplete data could be addressed, none of the reviewed articles attempted to resolve this issue [147, 148].

**4.1.3 Systematic method.** Many possible factors can be listed when determining the input and output variables. Nevertheless, this phenomenon can produce two significant issues: a lengthy input and output list and a negative impact on the DEA in accurately measuring efficiency if a limited number of DMUs are observed [139, 142]. Hence, selecting the significant variables and simultaneously accurately measuring efficiency is essential. The process has also been evaluated in various articles by incorporating systematic procedures. This review identified four systematic approaches: Delphi, Preference Ranking Organisation Method for Enrichment Evaluation (PROMETHEE), bibliometric analysis, and variance filter [48, 57, 60, 101, 108]. Consequently, these systematic approaches could formalise the judgmental process of stakeholder viewpoints (managers, economists, and policymakers). The Delphi method focuses on gathering the most reliable consensus of expert opinion for challenging situations. This forecasting method was initially presented in the 1950s by Olaf Helmer and Norman Dalkey of the Rand Corporation based on the responses from several iterations of questionnaires distributed to a panel of experts [149, 150]. Therefore, the Delphi method is widely recognised in assessing DEA efficiency in various areas [151–153].

The PROMETHEE method was initially developed in 1982 and underwent more advancements in 1985 [154–156]. This method is recognised as highly utilised and practical for multiple criteria decision aid (MCDA), including its application with DEA [157–160]. Compared to other MCDA methods, the PROMETHEE is considered a straightforward and computationally simple ranking system. The system incorporates weights indicating the relative

significance of each criterion alongside the preference function associated with each criterion. One of the critical applications of PROMETHEE involves its capability to assist decision-makers in choosing the optimal options for evaluating hospital performance. This process enables investigations to include PROMETHEE in DEA-based applications [161–164].

Pritchard is credited with coining the term "bibliometric" in 1969. This method is defined as an "application of mathematics and statistical methods to books and other media of communication". Therefore, bibliometric analysis evaluates the bibliographic information or metadata properties from a database or collection of documents to enhance understanding of the topic under investigation [165, 166]. Numerous articles have applied bibliometric analysis for various objectives as follows:

1. To identify new trends in journal performance, collaborative styles, and research components

2. To lay the groundwork for the new and significant advancements in a field

3. To systematically understand the massive amounts of unstructured data to interpret and map the cumulative scientific knowledge and evolutionary nuances of established domains [167, 168]

Hence, bibliometric analysis can determine the input and output variables in DEA studies involving the healthcare industry, such as hospitals.

The variance filter is a mechanism used in feature selection. The process determines and retains the most essential traits, helping to reduce noise, lower the computational expense of a model, and occasionally boost model performance. This review suggested that certain studies involved listing the crucial input and output variables [169, 170]. The variance filter (feature selection) allowed these studies to eliminate variables (input or output) with minimal or negligible impact on the efficiency measurement of DMUs. This method was also well accepted and commonly used in DEA-based articles [101, 171–173].

**4.1.4 Expert judgement.** Expert judgement can be utilised by researchers, stakeholders, or decision-makers to refine the input and output variable selections. A value judgement is "logical constructs used in efficiency assessment research that reflect the decision makers' preferences during the efficiency assessment procedure". This process includes the decision to exclude or assign a zero weight in the variable [142, 174], which depends on the efficiency measurement capacity of DEA based on the necessities and requirements of the decision-makers. Nonetheless, various constraints can develop, such as selection bias, input and output exclusions significantly impacting efficiency measurement or incorrect input and output weights. Therefore, researchers are encouraged to incorporate expert or value judgement to achieve their objectives. Different motivations are also observed for managers, economists, government policy makers, and academicians. Despite these individuals being committed to improving productivity, different judgements involving variable selections are presented. Considering that the DEA application possesses benefits and drawbacks, understanding the managerial and statistical implications of employing value judgment in input and output selections is crucial [175].

## 4.2 Managerial and economic implications in the input and output selection processes

This review empirically provided the approaches used in input and output variable selections for hospital evaluation-based DEA methods. The healthcare sector encounters daily challenges from public policy, resulting in new organisations, laws, and technology. Hence, managers

must address these concerns by implementing practical performance evaluation and decision-making strategies. These concerns necessitate careful performance and decision-making evaluations from economic and managerial perspectives. Generally, the input and output selections in DEA comprise two components: selected methods and variables. This selection process in the analysis can significantly impact the DEA outcomes. Conversely, this review indicated little consideration was devoted to the input and output variable selections in a real-world scenario. The DEA often owns an extensive initial list of potential variables to consider. Therefore, each resource that a DMU uses should be considered as an input variable. The assessment made by a manager or economist to justify the selection process also holds significant importance in practical situations.

The selection process in actual conditions is made more complex by the objective of production economics. These factors can include profit, quality control, and customer satisfaction. Hence, evaluating these several competing factors is a difficult task due to the influence of multiple decision-makers. For example, a profit-oriented DEA assessment can conflict with customer satisfaction. The results may not represent the production objective if the manager combines these measurements simultaneously. After analysing this review, certain judgements made from management and economic perspectives are proposed as follows:

1. Establish the objective production of the analysis, ensuring that all stakeholders easily understand it

2. Utilise the existing selection approaches to the greatest extent possible

3. Introduce a managerial-level committee to evaluate the variables before deciding on the final model

4. Physical units and managerial or economic perspectives differ according to the objective production of the analysis, such as comparing salary in dollars against the number of employees

### 4.3 Common DEA model parameters in hospital efficiency evaluation

**4.3.1 Model type.**   Significant advancements and transformations throughout time have been observed in the DEA application. Hence, numerous models are available to assess efficiency, ranging from general models to the more specialised use of DEA. Approximately 80.90% (72 of 89) of the examined articles in this review applied the radial DEA model. The models included were the BCC, CCR, and a combined BCC and CCR model. Consequently, these findings align with other healthcare-based reviews [17, 21, 33]. The focus of a radial DEA model is typically on the proportionate change in input or output values. Therefore, slacks (excess inputs or shortfall outputs still existing in the model) are ignored or treated as optional. Even though the radial model demonstrated various limitations, it is still commonly employed due to its fundamental nature, simplicity, and ease of application (minimal requirements on the production criteria of the DMUs) [47, 79, 176].

**4.3.2 Model orientation.**   Several factors or arguments can influence the DEA orientation selection, such as the decision maker's level of control, the nature of production, and the researcher's purpose from the model [25, 32, 177]. Healthcare organisations or hospitals generally possess limited or less control over their outputs. Nevertheless, this observation does not imply that a DEA efficiency evaluation in a hospital must be focused solely on input orientation. Thus, this review discovered that 55.06% (49 of 89) of the articles used input-oriented DEA models. Previous articles also highlighted similar findings with varying proportions

[17, 20, 21]. Researchers and hospital managers viewed reducing inputs while achieving a desired output level as a more appropriate measure of hospital efficiency. This outcome was attributed to the limited control hospitals possessed over their outputs.

**4.3.3 Returns to scale assumption.** Ongoing discussions have been observed concerning which of the two fundamental models (CRS or VRS) are superior. Hospital managers are actively searching for the most effective evaluation methods to assess the efficiency impact of various inputs and outputs on their organisations. Hence, selecting a return to scale involving a hospital is determined by the size of the hospital [64], organisation factors [75, 90], input and output process flow [110, 126], and technological involvement [81, 121]. Adopting an inappropriate return to scale can result in an excessively constrained search region for effective DMUs. Therefore, both assumptions should be examined to comprehend the implications of using either one [178]. This review discovered that most articles applied CRS and VRS assumptions for comparison (35.96%), followed by only VRS assumptions (32.58%). Previous articles also demonstrated a trend towards replacing CRS with VRS assumption in DEA-based applications [20, 21, 33]. Specifically, most hospital efficiency assessments focused on economies of scale and considered the non-proportional correlation between inputs and outputs in the healthcare production function.

**4.3.4 Input and output selections.** The methodologies involving input and output selections were effectively covered in this review, achieving the main objective of this study. Thus, appropriate input and output selections were crucial for the DEA analysis. Many previous articles denoted that the effectiveness of the DEA efficiency analysis was heavily influenced by the quality and quantity of these indicators [15–18, 20, 21, 30, 33]. Most articles (52.76%) used staff-related input as one of the variables in efficiency measurement. Given that human resources were significant in any organisation (including hospitals), this outcome was not surprising. Consequently, this finding was similar to previous DEA-related and performance-based articles on healthcare services [2, 13].

Typically, the analysis unit for staff-related factors is contingent upon the operational dynamics of the organisation. This review suggested two staff-related factors: the actual number of staff and the full-time equivalent. Various staff types were observed, from clinical to non-clinical (see S3 and S4 Appendices). The number of general beds was the highest (74.49%) when examining the sub-type of inputs, which hospital beds were a fundamental capital input for a hospital. This factor was a key indicator to assess hospital performance, capacity, and competency while comparing healthcare services across different countries [179–181]. Meanwhile, most articles applied production-related outputs rather than quality-related outputs. This phenomenon was attributed to the fact that it was simpler to quantify production-based data and provide stakeholders with a clear objective to improve upon it. Healthcare managers also would not prioritise effectiveness (quality) over efficiency [25, 182]. Overall, this review indicated that the common outputs applied were the number of inpatients, outpatients, and operations. Given that these outputs were the fundamental components of hospital services, the extensive utilisation of these factors was not unexpected.

**4.3.5 Extended analysis.** The classic DEA model is considered insufficient on its own because of the complexity of hospital processes and the continuous efforts of researchers and practitioners to enhance healthcare efficiency assessment. The majority of recent research studies on healthcare efficiency assessment integrate DEA with various approaches and techniques in order to address the weaknesses of the latter and offer a comprehensive and accurate picture of healthcare efficiency.

Forty extended analysis were observed within the reviewed articles. Despite the fact that each study had a different rationale for performing an extended analysis, a consistent theme was found.

1. To ascertain how contextual or environmental factors affect the efficiency scores [86, 108]

2. To quantitatively compare efficiency scores [52, 63]

3. To resolve the issues with serially linked estimates and produce bias-corrected efficiency estimates by utilising simulated distributions to compute the indices' standard errors and confidence ranges [82, 129]

4. To assess healthcare facilities' long-term performance using panel data analysis [76, 133]

5. To ascertain the relationship between the indicators that were to be included in the DEA model for input and output [45, 53]

6. To forecast, following the consideration of exogenous elements in the efficiency assessment, whether or not a healthcare unit should be deemed efficient [67, 132]

Consequently, in order to gain a better understanding of how these approaches were applied and helped the various researchers achieve their goals, it is necessary to recognise and credit these ways.

## 5. Limitations and conclusion

This novel systematic review represented the comprehensive investigation methods used to identify input-output variables for measuring hospital efficiency using DEA. To the authors' knowledge, no prior studies were conducted on this topic. The primary objective of this systematic review was to offer an overview of the existing approaches. This review also provided an update on the current application of DEA models for evaluating hospital efficiency. Approximately 89 articles were reviewed and assessed thoroughly with the specified objectives, and the literature review was primarily employed as a method for selecting inputs and output variables in DEA. These articles utilised literature review as a single method or combined with other approaches to enhance the robustness and vigour of the selection process. Considering that the selection of variables in DEA could lead to varying efficiency measurement outcomes, this process was considered crucial [139]. Nevertheless, no definitive approach or methodology could be identified for selecting variables (input-output) in DEA, concurrently representing its advantages and disadvantages [183–185].

Researchers and stakeholders should use the DEA to assess the effectiveness of their organisation according to their preferences. Conversely, these individuals should be aware of the limitations and potential constraints of DEA [139, 142]. Even though this review specifically examined methodologies employed in hospital settings, the scope of the findings could be restricted. Alternative procedures or methods could be utilised to select input and output variables for DEA studies in different fields or based on other perspectives [186]. Given that researchers and healthcare professionals aim to improve healthcare efficiency assessment, an optimal input-output selection approach should be identified. Hence, examining past, present, and potential developments in the DEA literature is essential due to its significant impact on DEA studies. The parameters for the DEA models also did not present any evidence to support an optimal or universally fitting model, for which almost all models were utilised multiple times (see S3 and S4 Appendices). Consequently, this review offered guidelines and methodological principles for conducting DEA studies based on established research. This process can provide insights to hospital managers, healthcare workers, policy officials, and students on the efficiency evaluation using DEA.

### 5.1 Registration and protocol

This study was registered at OSF Registries (https://osf.io/registries). All information regarding the registration and study protocol can be accessed at https://osf.io/nby9m or https://osf.io/e7mj9/?view_only=53deec8e6c6946eeaf0ea6fe2f0f212a.

## Supporting information

**S1 Appendix. PRISMA abstract checklist.**
(DOCX)

**S2 Appendix. PRISMA 2020 main checklist.**
(DOCX)

**S3 Appendix. Table 6 summary of 89 reviewed publications.**
(DOCX)

**S4 Appendix. Table 7 summary of 89 reviewed publications.**
(DOCX)

**S5 Appendix. Table 8 types of efficiency studied.**
(DOCX)

**S6 Appendix. Table 9 model types applied in the studies.**
(DOCX)

**S7 Appendix. Table 10 model orientation applied in the studies.**
(DOCX)

**S8 Appendix. Table 11 return to scale assumption applied in the studies.**
(DOCX)

## Author Contributions

**Conceptualization:** M. Zulfakhar Zubir, A. Azimatun Noor, A. M. Mohd Rizal, A. Aziz Harith, M. Ihsanuddin Abas.

**Data curation:** M. Zulfakhar Zubir, A. Azimatun Noor, A. M. Mohd Rizal, A. Aziz Harith, M. Ihsanuddin Abas.

**Formal analysis:** M. Zulfakhar Zubir, A. Azimatun Noor, A. M. Mohd Rizal, A. Aziz Harith, M. Ihsanuddin Abas.

**Funding acquisition:** M. Zulfakhar Zubir, A. Azimatun Noor, A. M. Mohd Rizal.

**Investigation:** M. Zulfakhar Zubir, A. Azimatun Noor, A. M. Mohd Rizal, A. Aziz Harith, M. Ihsanuddin Abas.

**Methodology:** M. Zulfakhar Zubir, A. Azimatun Noor, A. M. Mohd Rizal.

**Project administration:** M. Zulfakhar Zubir, A. Azimatun Noor, A. M. Mohd Rizal.

**Resources:** M. Zulfakhar Zubir, A. Azimatun Noor, A. M. Mohd Rizal.

**Software:** M. Zulfakhar Zubir.

**Supervision:** M. Zulfakhar Zubir, A. Azimatun Noor, A. M. Mohd Rizal.

**Validation:** M. Zulfakhar Zubir, A. Azimatun Noor, A. M. Mohd Rizal.

**Visualization:** M. Zulfakhar Zubir, A. Azimatun Noor, A. M. Mohd Rizal.

**Writing – original draft:** M. Zulfakhar Zubir, A. Azimatun Noor, A. M. Mohd Rizal, A. Aziz Harith, M. Ihsanuddin Abas, Zuriyati Zakaria, Anwar Fazal A. Bakar.

**Writing – review & editing:** M. Zulfakhar Zubir, A. Azimatun Noor, A. M. Mohd Rizal, A. Aziz Harith, M. Ihsanuddin Abas, Zuriyati Zakaria, Anwar Fazal A. Bakar.

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
