## [Decision Letter · Decision Letter 0]

5 Feb 2024

PONE-D-23-33407Approach in Inputs & Outputs Selection of Data Envelopment Analysis (DEA) Efficiency Measurement in Hospital: A Systematic ReviewPLOS ONE

Dear Dr. Aizuddin,

Thank you for submitting your manuscript to PLOS ONE. After careful consideration, we feel that it has merit but does not fully meet PLOS ONE’s publication criteria as it currently stands. Therefore, we invite you to submit a revised version of the manuscript that addresses the points raised during the review process.

We appreciate the opportunity to consider your work for publication. Your article has undergone a peer review process, and I have also conducted a thorough evaluation. After careful consideration, we have concluded that the manuscript requires major revisions before it can be considered for publication. This decision reflects a consensus between the external reviewer's insights and my own assessment as the second reviewer. The key areas that necessitate revision are as follows:Language and Clarity: There are several instances of typographical errors and language issues that need to be addressed. For example, in Section 3.2.3, the term "return to scale assumption" should be corrected to "returns to scale assumption". We recommend a thorough and careful revision of the English language used throughout the paper to ensure clarity and accuracy.Comparative Analysis from Managerial and Economic Viewpoints: The manuscript currently lacks a comparison of the reviewed papers from managerial and economic perspectives. Such an analysis is crucial for a comprehensive understanding of the subject matter. We advise you to incorporate a detailed comparative analysis that reflects these perspectives, thereby enriching the depth and relevance of your review. Please note that these revisions are substantial and critical to enhancing the quality and scholarly value of your paper.

We believe that addressing these issues will significantly improve the manuscript and better align it with the journal's standards and readership expectations and understand that these revisions may require considerable effort and time. However, we are confident that your dedication to these improvements will make a valuable contribution to the field.

We look forward to receiving your revised manuscript.

Kind regards,

André Ramalho, PhD

Academic Editor

PLOS ONE

Reviewers' comments:

Reviewer's Responses to Questions

**Comments to the Author**

1. Is the manuscript technically sound, and do the data support the conclusions?

Reviewer #1: Yes

2. Has the statistical analysis been performed appropriately and rigorously? 

Reviewer #1: Yes

3. Have the authors made all data underlying the findings in their manuscript fully available?

Reviewer #1: Yes

4. Is the manuscript presented in an intelligible fashion and written in standard English?

Reviewer #1: Yes

5. Review Comments to the Author

Reviewer #1: Dear Editor,

Thank you for giving me the opportunity to review the paper entitled "Approach in Inputs & Outputs Selection of Data Envelopment Analysis (DEA) Efficiency Measurement in Hospital: A Systematic Review" (with Manuscript Number: PONE-D-23-33407).

I have completed reading this paper. It is a interesting review paper. However, based on the following comments, I believe the current version of this paper is not appropriate for publication in "PLOS ONE".

Thus, I recommend major revision for this paper.

Yours sincerely,

Comments:

1. There are several dictations errors in the text. For instance, in the title of Section 3.2.3, the term "return to scale assumption" must be modified to the term "returns to scale assumption". English language of this paper must be revised carefully.

2. The authors have not compared the papers reviewed in this paper from managerial and economic viewpoints.

6. PLOS authors have the option to publish the peer review history of their article (what does this mean?). If published, this will include your full peer review and any attached files.

Reviewer #1: No

---

## [Author Response · Author response to Decision Letter 0]

2 May 2024

André Ramalho, PhD

Academic Editor

PLOS ONE

San Francisco, California, USA

1st May 2024

Dear Dr. André Ramalho: 

Thank you for inviting us to submit a revised draft of our manuscript entitled, "Approach in Inputs & Outputs Selection of Data Envelopment Analysis (DEA) Efficiency Measurement in Hospital: A

Systematic Review" to PLOS ONE. We also appreciate the time and effort you and each of the reviewer have dedicated to providing insightful feedback on ways to strengthen our paper. Thus, it is with great pleasure that we resubmit our article for further consideration. We have incorporated changes that reflect the detailed suggestions you have graciously provided. We also hope that our edits and the responses we provide below satisfactorily address all the issues and concerns you and the reviewer have noted.

To facilitate your review of our revisions, the following is a point-by-point response to the questions and comments delivered in your letter dated 5th Feb 2024 [PONE-D-23-33407]. The responses to the concerns raised by editor and reviewer are below and are color coded as follows: a) Comments from editors or reviewers are shown as [text]; b) Our responses are shown as [text].

The comments were very helpful overall, and we are appreciative of such constructive feedback on our original submission. After addressing the issues raised, we feel the quality of the paper is much improved.

EDITOR SUGGESTIONS:

1. [We appreciate the opportunity to consider your work for publication. Your article has undergone a peer review process, and I have also conducted a thorough evaluation. After careful consideration, we have concluded that the manuscript requires major revisions before it can be considered for publication. This decision reflects a consensus between the external reviewer's insights and my own assessment as the second reviewer. The key areas that necessitate revision are as follows:]

• RESPONSE: [We wish to thank you and the reviewer for the assessment and consideration for publication. We agree with you and did major revisions throughout our paper. In our revisions, we have attempted to address key areas that necessitate revision. Details of the changes is the document “Revised Manuscript with Track Changes”. However, we have retained the structure of the paper to cover broad reach of audience including those who are new to Data Envelopment Analysis]

2. [Language and Clarity: There are several instances of typographical errors and language issues that need to be addressed. For example, in Section 3.2.3, the term "return to scale assumption" should be corrected to "returns to scale assumption". We recommend a thorough and careful revision of the English language used throughout the paper to ensure clarity and accuracy.]

• RESPONSE: [We thank Editor for this comment. We regret for the typographical errors and language issues. Accordingly, we have revised, edited and proofread the whole paper to improve the clarity and accuracy. We have also replaced the pointed-out term and others to be more in line with your comments. We hope that the edited paper clarifies this comment]

3. [Comparative Analysis from Managerial and Economic Viewpoints: The manuscript currently lacks a comparison of the reviewed papers from managerial and economic perspectives. Such an analysis is crucial for a comprehensive understanding of the subject matter. We advise you to incorporate a detailed comparative analysis that reflects these perspectives, thereby enriching the depth and relevance of your review. Please note that these revisions are substantial and critical to enhancing the quality and scholarly value of your paper.]

• RESPONSE: [We appreciate your insightful and fair comments. We agreed that the paper lacks in the areas mentioned. However, we believed that the paper should interest broad reach of audience both in and outside of DEA discipline. This includes student and those who a new to DEA. Therefore, we incorporated the managerial and economic analysis by adding crucial statements and specific section to reflects the perspective] [Pages and lines referred to manuscript with All Markup] [ The added analysis is highlighted in yellow in the revised manuscript.]

(i) P.15, lines 353-354 – We elaborated the checklist used was specific to assess from economic perspective

(ii) P.38, lines 810-815 – We explained the method mentioned acceptable for academic or research purpose. But as manager or economic purpose the method need to be use wisely.

(iii) P.40, lines 863-865 – We added the statement that, the method stated can formalized the judgmental process. This is important from managerial or economic perspective as they need a concrete base or definitive stand in their decision.

(iv) P.42-43, lines 937-943 – We elaborated the motivation of manager or economist are different. This motivation in managerial and economic area will change their judgement.

• [We added a specific section to enrich the paper as suggested and at the same time keep the paper structure to attract more reader especially those who are new to DEA. P.43-45, lines 951-1005, point 4.2 – Specific section to the main objective of the paper. We expanded the discussion from managerial and economic standpoints and proposed specific recommendation. Again, although this recommendation reflects to the specified group it is also relevance to others who is interested.]

• [We hope these revisions provide a more balanced and thorough discussion in the paper.]

REVIEWER 1 COMMENTS:

4. [Thank you for giving me the opportunity to review the paper entitled "Approach in Inputs & Outputs Selection of Data Envelopment Analysis (DEA) Efficiency Measurement in Hospital: A Systematic Review" (with Manuscript Number: PONE-D-23-33407).

I have completed reading this paper. It is a interesting review paper. However, based on the following comments, I believe the current version of this paper is not appropriate for publication in "PLOS ONE".

Thus, I recommend major revision for this paper]

• RESPONSE: [We wish to thank you for your insightful comments and pleased to find that this paper interest you. We revised the paper as you suggested in the document “Revised Manuscript with Track Changes”. Your comments and suggestion have greatly helped us to improve the quality of our paper.]

5. [There are several dictations errors in the text. For instance, in the title of Section 3.2.3, the term "return to scale assumption" must be modified to the term "returns to scale assumption". English language of this paper must be revised carefully.]

• RESPONSE: [We thank you for noting this. We are very sorry for this dictations error and for the English language was somewhat inadequate. Thus, we reviewed back the paper and improved in term of language, structure and related term on DEA. Proofreading was done accordingly to make the paper better. We hope the revised paper addressed your concern]

6. [The authors have not compared the papers reviewed in this paper from managerial and economic viewpoints.]

• RESPONSE: [Thank you for providing these insights. We agreed the paper lacks from the viewpoints mentioned. We also believed it is more appropriate for the paper to be more general as to attract larger and broader audience. Thus, we decided to incorporated the managerial and economic analysis by adding crucial statements and specific section to reflects the viewpoints. Our response is the same as what we replied to the Editor.] [Pages and lines referred to manuscript with All Markup] [ The added analysis is highlighted in yellow in the revised manuscript.]

(i) P.15, lines 353-354 – We elaborated the checklist used was specific to assess from economic perspective

(ii) P.38, lines 810-815 – We explained the method mentioned acceptable for academic or research purpose. But as manager or economic purpose the method need to be use wisely.

(iii) P.40, lines 863-865 – We added the statement that, the method stated can formalized the judgmental process. This is important from managerial or economic perspective as they need a concrete base or definitive stand in their decision.

(iv) P.42-43, lines 937-943 – We elaborated the motivation of manager or economist are different. This motivation in managerial and economic area will change their judgement.

• [We added a specific section to enrich the paper as suggested and at the same time keep the paper structure to attract more reader especially those who are new to DEA. P.43-45, lines 951-1005, point 4.2 – Specific section to the main objective of the paper. We expanded the discussion from managerial and economic standpoints and proposed specific recommendation. Again, although this recommendation reflects to the specified group it is also relevance to others who is interested.]

• [We hope that these revisions are sufficient to make our paper suitable for publication.]

CONCLUDING REMARKS: Thank you for providing us the opportunity to improve our manuscript with your useful suggestions and questions. We worked hard to incorporate your feedback and hope that these changes would persuade you to accept our submission.

Sincerely,

[A Azimatun Noor]

Corresponding Author:

A Azimatun Noor

Associate Prof Dr

Universiti Kebangsaan Malaysia

[Faculty of Medicine UKM

Level 2, Kompleks Pendidikan Perubatan Canselor Tuanku Jaafar

Jalan Yaacob Latif, Bandar Tun Razak

56000 Cheras, Kuala Lumpur

MALAYSIA]

[azimatunnoor@ppukm.ukm.edu.my]

[Tel: +603 – 9145 8693]

[Fax: +603 – 9145 8401]

Additional Contact:

M Zulfakhar Zubir

Dr

Institution/Affiliation Name

[Medical Development Division, Ministry of Health Malaysia, Putrajaya, MALAYSIA]

[dr.zulfakhar@moh.gov.my]

[Tel: +6019 – 6686415]

---

## [Decision Letter · Decision Letter 1]

21 May 2024

PONE-D-23-33407R1Approach in Inputs & Outputs Selection of Data Envelopment Analysis (DEA) Efficiency Measurement in Hospitals: A Systematic ReviewPLOS ONE

Dear Dr. Aizuddin,

Thank you for submitting your manuscript to PLOS ONE. After careful consideration, we feel that it has merit but does not fully meet PLOS ONE’s publication criteria as it currently stands. Therefore, we invite you to submit a revised version of the manuscript that addresses the points raised during the review process.

We look forward to receiving your revised manuscript.

Kind regards,

André Ramalho, PhD

Academic Editor

PLOS ONE

Journal Requirements:

Reviewers' comments:

Reviewer's Responses to Questions

**Comments to the Author**

1. If the authors have adequately addressed your comments raised in a previous round of review and you feel that this manuscript is now acceptable for publication, you may indicate that here to bypass the “Comments to the Author” section, enter your conflict of interest statement in the “Confidential to Editor” section, and submit your "Accept" recommendation.

Reviewer #1: All comments have been addressed

Reviewer #2: (No Response)

2. Is the manuscript technically sound, and do the data support the conclusions?

Reviewer #1: Yes

Reviewer #2: Partly

3. Has the statistical analysis been performed appropriately and rigorously? 

Reviewer #1: Yes

Reviewer #2: N/A

4. Have the authors made all data underlying the findings in their manuscript fully available?

Reviewer #1: Yes

Reviewer #2: Yes

5. Is the manuscript presented in an intelligible fashion and written in standard English?

Reviewer #1: Yes

Reviewer #2: Yes

6. Review Comments to the Author

Reviewer #1: Dear Editor,

Thank you for giving me the opportunity to review the revised paper entitled "Approach in Inputs & Outputs Selection of Data Envelopment Analysis (DEA) Efficiency Measurement in Hospitals: A Systematic Review" (with Manuscript Number: PONE-D-23-33407R1).

I have completed reading the revised paper. In my opinion, the authors have addressed the points raised in the previous review with patient.

I believe that the revised paper is now acceptable for publication in "PLOS ONE".

Yours sincerely,

Reviewer #2: Thank you for the opportunity of reviewing this work. Authors conducted a systematic review on the use of DEA for performing hospital efficiency assessment, focusing on input-output selection. This review touches a very relevant topic in health economics. Authors have assessed a considered number of publications, summarizing and describing key methodological aspects behind the use of DEA for hospital efficiency assessment. Overall, the issue is relevant, and the review is worth for publication, but some major improvements need to be done. Below I provide comments and suggestions separated by sections:

Introduction

Line 66: "[...] and a framework is designed. in keeping with the analysis. [...]" -> It seems there is text missing in this sentence.

Line 133: " [...] Despite the large number of studies on the use of DEA in hospitals, efforts to comprehensively examine these studies are still missing and warrant more investigation. This article identifies and describes DEA research in hospitals in an effort to close the knowledge gap. [...]" -> I am afraid this may not be entirely true. There are several published review articles that have assessed and described the use of DEA in the hospital sector, including all methodological aspects within this methodology, such as input-output selection. Having said that, the motivation and need for this review should be clarified and authors need to clearly state what the review adds to other existing and similar reviews.

Line 143: "In addition, this study observed the current trend in analysing hospital efficiency using DEA" -> What does this objective means exactly? It does not seem that this analysis was performed in the Results sections. Could you please clarify this point?

Methods

Authors used 6 standard, comprehensive databases and justified their choice. However, Methods section lacks description on the search strategy and query. It was only mentioned the terms used. It is important to present at least the final query used for searching the databases. Also, it is not completely clear how many reviewers participated in each phase (abstract screening, full-text analysis and data extraction). Authors only mention 2 readers and then a third one for consensus in the Quality Appraisal section. Authors need to clarify how the review process was carried out. Furthermore, inclusion and exclusion criteria considered for abstract screening and full-text selection are not explicitly mentioned in the Methods section. This information is usually basic in systematic reviews. Authors only mentioned type of publication, language and period criteria.

Results

I think Tables 1 and 2 need to be restructured. It would be easier to read if Table 1 contains only the full list of input variables and Table 2 only the full list of output variables (including categories and counts), for instance. Moreover, it is important to include the definitions and units of the variables in these tables.

The section "3.2.5 Extended analysis & data type" only lists a series of additional methods employed but provide little to no information on the purpose or usefulness of applying these extended analyses. For instance, some of the methods presented are typically employed in second-stage analysis (e.g., Tobit regression) to explain the efficiency or inefficiency levels found by DEA. Without this information, I do not see what this section adds.

Discussion and conclusions

Authors provided an in-depth discussion on input-output selection, highlighting the existing variability concerning the selection of inputs and outputs. Based on the presented results, it seems that input-output selection is mostly influenced by data availability and research scope, and thus it is not a methodological preference, making it difficult to suggest common practices. However, I missed the main message to be conveyed by the review. I was expecting to find more concrete recommendations and guidelines regarding input-output selection in the Discussion and Conclusion sections, or at least a more directed discussion on how variable selection affects efficiency estimation or what key considerations for variable selection should be addressed.

Finally, it is important that authors carefully review the text in order to improve clarity and English.

7. PLOS authors have the option to publish the peer review history of their article (what does this mean?). If published, this will include your full peer review and any attached files.

Reviewer #1: **Yes: **Mohammad Khoveyni

Reviewer #2: No

---

## [Author Response · Author response to Decision Letter 1]

10 Jun 2024

Response to Reviewers:

André Ramalho, PhD

Academic Editor

PLOS ONE

San Francisco, California, USA

5th June 2024

Dear Dr. André Ramalho: 

Thank you for inviting us to submit a revised version of our manuscript entitled, "Approach in Inputs & Outputs Selection of Data Envelopment Analysis (DEA) Efficiency Measurement in Hospital: A Systematic Review" to PLOS ONE. We also appreciate the time and effort you and each of the reviewers have dedicated to providing insightful feedback on ways to strengthen our paper. Thus, it is with great pleasure that we resubmit our article for further consideration. We have incorporated changes that reflect the detailed suggestions you have graciously provided. We also hope that our edits and the responses we provide below satisfactorily address all the issues and concerns you and the reviewer have noted.

To facilitate your review of our revisions, the following is a point-by-point response to the questions and comments delivered in your letter dated 21st May 2024 [PONE-D-23-33407R1]. The responses to the concerns raised by editor and reviewers are below and are color coded as follows: a) Comments from editors or reviewers are shown as [text]; b) Our responses are shown as [text].

The comments were very helpful overall, and we are appreciative of such constructive feedback on our original submission. After addressing the issues raised, we feel the quality of the paper is much improved.

EDITOR SUGGESTIONS:

1. [Thank you for submitting your manuscript to PLOS ONE. After careful consideration, we feel that it has merit but does not fully meet PLOS ONE’s publication criteria as it currently stands. Therefore, we invite you to submit a revised version of the manuscript that addresses the points raised during the review process. Minor Revision]

• RESPONSE: [We are really glad that the article reaches you. We also appreciate the time and effort you and each of the reviewer have dedicated to providing insightful feedback on ways to strengthen our paper. Previously the article needs a major revision, we are delightful that the second revision require minor revision. Thus, it is with great pleasure that we resubmit our article for further consideration. We have incorporated changes that reflect the detailed suggestions you have graciously provided. We also hope that our edits and the responses we provide below satisfactorily address all the issues and concerns you and the reviewers have noted.]

• [We appreciate both of the reviewers insightful and fair comments. Thank you for Reviewer #1 for the recommendation and acceptance for publication. We agreed with Reviewer #2 for the additional comment and suggestion. Therefore, we incorporated the concerns raised by Reviewer #2] [Pages and lines referred to manuscript with All Markup] [ The added content or response to comment is highlighted in yellow in the revised manuscript.]

• [These changes, we hope, will provide the paper a more comprehensive and fair discussion.]

REVIEWER 1 COMMENTS:

[Dear Editor,

Thank you for giving me the opportunity to review the revised paper entitled "Approach in Inputs & Outputs Selection of Data Envelopment Analysis (DEA) Efficiency Measurement in Hospitals: A Systematic Review" (with Manuscript Number: PONE-D-23-33407R1).

I have completed reading the revised paper. In my opinion, the authors have addressed the points raised in the previous review with patient.

I believe that the revised paper is now acceptable for publication in "PLOS ONE".

Yours sincerely,]

• RESPONSE: [We would like to thank you for taking the necessary time and effort to review the article. We sincerely appreciate all your valuable comments and suggestions, which helped us in improving the quality of the article. We would like to take this opportunity to thank you for the effort and expertise that you contributed towards reviewing the article, without which it would be impossible to maintain the high standards of peer-reviewed journals. Thank you for considering the article acceptable for publication in “PLOS ONE”.]

REVIEWER 2 COMMENTS:

1. [Thank you for the opportunity of reviewing this work. Authors conducted a systematic review on the use of DEA for performing hospital efficiency assessment, focusing on input-output selection. This review touches a very relevant topic in health economics. Authors have assessed a considered number of publications, summarizing and describing key methodological aspects behind the use of DEA for hospital efficiency assessment. Overall, the issue is relevant, and the review is worth for publication, but some major improvements need to be done. Below I provide comments and suggestions separated by sections:]

• RESPONSE: [We wish to thank you for your insightful comments and pleased to find that this paper interest you. We revised the paper as you suggested in the document “Revised Manuscript with Track Changes”. Your comments and suggestion have greatly helped us to improve the quality of our paper.]

• [ The added content or response to comment is highlighted in yellow in the revised manuscript.]

• [We hope these revisions provide a more balanced and thorough discussion in the paper.]

2. [Introduction

Line 66: "[...] and a framework is designed. in keeping with the analysis. [...]" -> It seems there is text missing in this sentence.

Line 133: " [...] Despite the large number of studies on the use of DEA in hospitals, efforts to comprehensively examine these studies are still missing and warrant more investigation. This article identifies and describes DEA research in hospitals in an effort to close the knowledge gap. [...]" -> I am afraid this may not be entirely true. There are several published review articles that have assessed and described the use of DEA in the hospital sector, including all methodological aspects within this methodology, such as input-output selection. Having said that, the motivation and need for this review should be clarified and authors need to clearly state what the review adds to other existing and similar reviews.

Line 143: "In addition, this study observed the current trend in analysing hospital efficiency using DEA" -> What does this objective means exactly? It does not seem that this analysis was performed in the Results sections. Could you please clarify this point?]

• RESPONSE: [We thank you for noting this. Some of the comments were already changed in the first review. We will point out the comment based on the line you mentioned.]

• Line 66: This statement was changed in the manuscript.

o Line 66: A framework is required following the analysis process. The optimal approach to implementing performance measurement is not to identify a minor adjustment as a supporting role to enhance one aspect of the health system outcomes. Instead, this identification should be utilised as a general strategy in gauging performance among the various system components.

• Line 133: This statement was changed to reflect and clarify the motivation for the produce manuscript. We thank you for mentioning this.

o Line 125: Although numerous hospital-based DEA articles have been recorded, inadequate complete analysis has been observed. Consequently, this outcome requires further investigation, leading to a research gap involving hospital-based DEA articles.

• Line 143: We revised this statement to explain that the review addressed the research gap mentioned. What we meant by the current trend is that we looked at few findings in previous review and compare briefly in the Discussion section.

o Line 131: To the authors' knowledge, no reviews regarding hospital-based DEA articles involving optimal input and output variable selections were reported. Thus, this review addressed this research gap by observing the current trends in hospital-based DEA analyses.

o Line 636: Consequently, these findings align with other healthcare-based reviews [17,21,33].

o Line 651: Previous articles also highlighted similar findings with varying proportions [17,20,21].

o Line 667: Previous articles also demonstrated a trend towards replacing CRS with VRS assumption in DEA-based applications [20,21,33].

o Line 681: Consequently, this finding was similar to previous DEA-related and performance-based articles on healthcare services [180,181].

3. [Methods

Authors used 6 standard, comprehensive databases and justified their choice. However, Methods section lacks description on the search strategy and query. It was only mentioned the terms used. It is important to present at least the final query used for searching the databases. Also, it is not completely clear how many reviewers participated in each phase (abstract screening, full-text analysis and data extraction). Authors only mention 2 readers and then a third one for consensus in the Quality Appraisal section. Authors need to clarify how the review process was carried out. Furthermore, inclusion and exclusion criteria considered for abstract screening and full-text selection are not explicitly mentioned in the Methods section. This information is usually basic in systematic reviews. Authors only mentioned type of publication, language and period criteria.]

• RESPONSE: [Thank you for providing these insights. We agreed the paper lacks from the viewpoints mentioned. Some of the comments were already addressed in the first review.]

• We added the final query for the searching database. General query string is added as the six databases have different search setting and requirement.

• We added in information on reviewers and method involve in each phase.

i. Line 179: The records were exported from the databases into Microsoft Excel sheet for screening. The final query string is as follow:

ii. Line 186: The titles and abstracts were independently screened by three reviewers.

iii. Line 228: Four reviewers extracted the data independently using a standardized data extraction form which is organized using Microsoft Excel.

• The review process was also explained in the Quality Appraisal section

i. Line 216: Thus, the scientific soundness of the chosen article was investigated using two tools to improve robustness and minimise bias. Two co-authors from different institutions evaluated each selected article separately using both tools to enhance reliability. A third reviewer was then requested to assess an article if a disagreement occurred.

ii. We believe this statement is adequate in explaining the process

• The inclusion and exclusion criteria were mentioned in the manuscript and in the PRISMA diagram.

i. Line 187 to 191 explained in details with regards to this matter.

ii. Line 194: PRISMA diagram further explained the process for final articles to be included in the review.

iii. We believe this will give better flow to audience in reading the manuscript.

4. [Results

I think Tables 1 and 2 need to be restructured. It would be easier to read if Table 1 contains only the full list of input variables and Table 2 only the full list of output variables (including categories and counts), for instance. Moreover, it is important to include the definitions and units of the variables in these tables. 

The section "3.2.5 Extended analysis & data type" only lists a series of additional methods employed but provide little to no information on the purpose or usefulness of applying these extended analyses. For instance, some of the methods presented are typically employed in second-stage analysis (e.g., Tobit regression) to explain the efficiency or inefficiency levels found by DEA. Without this information, I do not see what this section adds.]

• RESPONSE: [Thank you and we appreciate on the suggestion for the table structure. Table 1 are summary for both input and output and it is presented in aggregated information. We believe this is appropriate for those who are well verse in DEA for quick summary. While Table 2 and 3 are details that explained Table 1. The table 2 and 3 are in raw data which include the self-explained definitions and units.]

o For example, Table 1 for “Beds” is general term, Table 2 describe the type of beds, and units of measurement.

• Thank you for providing these insights. We agreed the paper lacks from the viewpoints mentioned "3.2.5 Extended analysis & data type". Thus, we elaborated this area in the discussion and at the same time we have mentioned the important of this section briefly in previous revision. The added content or response to comment is highlighted in yellow in the revised manuscript.

o Line 76: Although the theoretical and methodological limitations have been acknowledged in DEA, this method has attracted interest from researchers who aim to address the limitations. Hence, these studies have developed multiple methods integrating DEA with other statistical techniques and methodologies to improve efficiency evaluation.

o Line 121: and development of novel knowledge and approaches concerning DEA assessment.

o Line 284: This model could be further analysed or extended through a second stage or integrated with other statistical methods. Consequently, this process could improve efficiency measurement, understanding of the variation or difference in organisational performance, and evaluation of the productivity of the organisation over a specific period.

o Line 697 – 717: New section “4.2.5 Extended Analysis”

5. [Discussion and conclusions

Authors provided an in-depth discussion on input-output selection, highlighting the existing variability concerning the selection of inputs and outputs. Based on the presented results, it seems that input-output selection is mostly influenced by data availability and research scope, and thus it is not a methodological preference, making it difficult to suggest common practices. However, I missed the main message to be conveyed by the review. I was expecting to find more concrete recommendations and guidelines regarding input-output selection in the Discussion and Conclusion sections, or at least a more directed discussion on how variable selection affects efficiency estimation or what key considerations for variable selection should be addressed.

Finally, it is important that authors carefully review the text in order to improve clarity and English.]

• RESPONSE: [We appreciate the comment on this. We added this section during the first revision. Section “4.2 Managerial and economic implications in the input and output selection processes” explained in detail with regards to the review. We outlined the recommendations and guidelines in input-output selection.]

• Furthermore, we also remark few statements in the review to reflects on these perspectives

o Line 208: This checklist assessed the article from an economic perspective to ensure the findings could be used in policy analysis and managerial decisions.

o Line 509: Even though this method remained valid for academicians, other assessors (managers, economists or policymakers) could perceive it as contradictory to their practical perspectives. Hence, several factors must be considered from these individuals’ perspectives, including different indicators, production objectives, and policies.

o Line 541: Consequently, these systematic approaches could formalise the judgmental process of stakeholder viewpoints (managers, economists, and policymakers).

o Line 587: Therefore, researchers are encouraged to incorporate expert or value judgement to achieve their objectives. Different motivations are also observed for managers, economists, government policy makers, and academicians. Despite these individuals being committed to improving productivity, different judgements involving variable selections are presented.

• However, we believed that the paper should interest broad reach of audience both in and outside of DEA discipline. This includes student and those who a new to DEA. Therefore, this added section was written with view to be read and understand by audience regardless of their DEA knowledge background. 

• We reviewed back the paper and improved in term of language, structure and related term on DEA. Proofreading was done accordingly to make the paper better. We include the certificate of Proofreading in this revised manuscript. We hope the revised paper addressed your concern

CONCLUDING REMARKS: We appreciate y

---

## [Decision Letter · Decision Letter 2]

27 Jun 2024

Approach in Inputs & Outputs Selection of Data Envelopment Analysis (DEA) Efficiency Measurement in Hospitals: A Systematic Review

PONE-D-23-33407R2

Dear Dr. Aizuddin,

We’re pleased to inform you that your manuscript has been judged scientifically suitable for publication and will be formally accepted for publication once it meets all outstanding technical requirements.

Kind regards,

André Ramalho, PhD

Academic Editor

PLOS ONE

Additional Editor Comments (optional):

Reviewers' comments:

Reviewer's Responses to Questions

**Comments to the Author**

1. If the authors have adequately addressed your comments raised in a previous round of review and you feel that this manuscript is now acceptable for publication, you may indicate that here to bypass the “Comments to the Author” section, enter your conflict of interest statement in the “Confidential to Editor” section, and submit your "Accept" recommendation.

Reviewer #2: All comments have been addressed

2. Is the manuscript technically sound, and do the data support the conclusions?

Reviewer #2: Yes

3. Has the statistical analysis been performed appropriately and rigorously? 

Reviewer #2: N/A

4. Have the authors made all data underlying the findings in their manuscript fully available?

Reviewer #2: Yes

5. Is the manuscript presented in an intelligible fashion and written in standard English?

Reviewer #2: Yes

6. Review Comments to the Author

Reviewer #2: Dear Editor,

Thank you very much for the opportunity to review the paper entitled "Approach in Inputs & Outputs Selection of Data Envelopment Analysis (DEA) Efficiency Measurement in Hospitals: A Systematic Review" (PONE-D-23-33407R1).

I have read the revised content and the authors have carefully addressed all comments made the previous review. The revised version has been substantially improved and all key queries have been solved.

Therefore, I believe that the revised paper is now acceptable for publication in "PLOS ONE".

Kind regards.

7. PLOS authors have the option to publish the peer review history of their article (what does this mean?). If published, this will include your full peer review and any attached files.

Reviewer #2: No

---

## [Editor Report · Acceptance letter]

4 Jul 2024

PONE-D-23-33407R2 

PLOS ONE

Dear Dr. Noor, 

I'm pleased to inform you that your manuscript has been deemed suitable for publication in PLOS ONE. Congratulations! Your manuscript is now being handed over to our production team.

Kind regards, 

on behalf of

Prof. Dr. André Ramalho 

Academic Editor

PLOS ONE